# Production of zosteric acid and other sulfated phenolic biochemicals in microbial cell factories

Christian Bille Jendresen [1,2] & Alex Toftgaard Nielsen [1,2]

Biological production and application of a range of organic compounds is hindered by their limited solubility and toxicity. This work describes a process for functionalization of phenolic compounds that increases solubility and decreases toxicity. We achieve this by screening a wide range of sulfotransferases for their activity towards a range of compounds, including the antioxidant resveratrol. We demonstrate how to engineer cell factories for efficiently creating sulfate esters of phenolic compounds through the use of sulfotransferases and by optimization of sulfate uptake and sulfate nucleotide pathways leading to the 3'-phosphoadenosine 5'-phosphosulfate precursor (PAPS). As an example we produce the antifouling agent zosteric acid, which is the sulfate ester of $p$-coumaric acid, reaching a titer of 5 g L$^{-1}$ in fed-batch fermentation. The described approach enables production of sulfate esters that are expected to provide new properties and functionalities to a wide range of application areas.

[1] The Novo Nordisk Foundation Center for Biosustainability, Technical University of Denmark, Kemitorvet Building 220, 2800 Kgs Lyngby, Denmark. [2] Cysbio ApS, Agern Allé 1, 2970 Hørsholm, Denmark. Correspondence and requests for materials should be addressed to C.B.J. (email: cbj@cysbio.com) or to A.T.N. (email: atn@biosustain.dtu.dk)

Sulfate residues provide distinct properties to organic molecules, increasing negative charge, acidity and solubility, which may have biotechnological applications. Small phenolic compounds are naturally sulfated (O-sulfonated) in vivo; in plants controlling signaling[1], developmental regulation[2], plant–bacteria communication[3], and stress responses[4], and similarly in animals, steroid hormones are subject to sulfation and desulfation, altering activity, life-span, transport properties of the molecules[5], while xenobiotics are detoxified by sulfation. Several natural polymers are also subject to sulfation, having a role in interactions, disease progression, biomineralization, virulence, membrane integrity and osmoprotection. From a biotechnological perspective, the alterations of the physical and chemical properties of a compound through sulfation may have direct application on drugs and small molecule administration, and the properties may also have application in synthesis of novel polymers from sulfated monomers. The altered properties may also provide a means to produce an improved form of phenolic compounds that are generally poorly soluble and toxic to the production host.

Eelgrass (Zostera species) are marine plants that produce a naturally sulfated small compound, zosteric acid, the sulfate ester of p-coumaric acid. Extracts of Zostera have provided complex mixtures of compounds with a large fraction being zosteric acid[6,7], which have been shown to prevent biofouling by preventing the attachment of bacteria, yeast, fungi, algae and bivalves to solid surfaces or plant leafs[8]. Application of zosteric acid as a natural anti-adhesive agent may replace other chemical treatments in e.g. agriculture and for industrial and medical applications.

Zostera marina and Zostera noltii are prominent natural producers of zosteric acid, and while they have an important role in marine environments, plant material washed up on beaches have been proposed to be a source of natural zosteric acid[6]. However, this is seasonally dependent, requires extraction and may be limited by insufficient content or availability of marine plant material. Biological production of plant chemicals through genetic engineering of well-known microorganisms may be a preferred alternative for controlled production.

Zosteric acid may be formed chemically by sulfation (O-sulfonation) of p-coumaric acid, but a biological process would eliminate harsh chemical conditions and reduce waste generation. Here, we have manipulated Escherichia coli and Saccharomyces cerevisiae to produce zosteric acid from p-coumaric acid, by the action of a 3′-phosphoadenosine 5′-phosphosulfate (PAPS)-dependent sulfotransferase (Fig. 1). Coupling to a tyrosine ammonia-lyase (TAL) activity[9], p-coumaric acid is formed from tyrosine by non-oxidative deamination and subsequently converted to zosteric acid. By optimizing the uptake and activation of sulfate, we were able to enhance the production of zosteric acid from either p-coumaric acid, tyrosine or glucose. We additionally show that the sulfation activity is also applicable towards a large number of other phenolic compounds, including flavonoids, polyphenols, monomers for polymers and drugs, depending on the choice of sulfotransferase.

## Results

### Sulfation of phenolics in E. coli using sulfotransferases.
Although natural sulfate conjugated phenolic compounds offer interesting biological properties, they have been subject to only little biotechnological investigation[10]. Zosteric acid has attracted attention based on its abundance in Zostera marina and its potency for inhibiting biofilm formation. While zosteric acid is a naturally occurring product in eelgrass, its native biosynthesis pathway remains to be characterized. Zosteric acid could be formed by the formation of a sulfate ester of p-coumaric acid catalyzed by 3′-phosphoadenosine 5′-phosphosulfate (PAPS)-dependent aryl sulfotransferases.

The genomic sequence of the zosteric acid-producing plant Zostera marina has recently been published[11], and while it was noted that the genome harbors 12 genes putatively encoding arylsulfotransferases, substrate prediction of these enzymes is challenging[12], and it was suggested that some of these might be responsible for carbohydrate sulfation. Possibly, one of these genes could confer the activity required for specific production of zosteric acid, also if employed in a microbial cell factory. Since PAPS is part of the sulfate assimilation and reduction pathways across all kingdoms of life, a microbial cell factory should be able to utilize its native PAPS supply for the process. Thus, we examined the proposed arylsulfotransferases from Zostera marina, as well as a stress-inducible sulfotransferase from Arabidopsis thaliana[4,13] by heterologously expressing codon-optimized genes in E. coli, a preferred host for production of biochemicals. We used a KRX strain, for cloning and tight regulation of gene expression from T7 promoters through induction of T7 RNA polymerase by IPTG and rhamnose (Supplementary Fig. 1). The strain combines the T7 RNA polymerase under control of a rhamnose promoter that enables efficient protein overexpression in the absence of recombination and nuclease activities. The genes were cloned after a T7 promoter and lac operator sites onto a plasmid vector also expressing lacI. However, we did not detect zosteric acid upon expression of any of the plant genes (Fig. 2a).

The enzymatic activity required for producing compounds such as zosteric acid may, however, also be found in other organisms. Phenolic compounds, which are often toxic for animals, are subject to phase II detoxification, especially in the liver, where a general mechanism results in the formation of a less toxic conjugated version of the phenolic compound[14]. One of these reactions result in the formation of a sulfate ester, catalyzed by PAPS-dependent aryl sulfotransferases. We therefore cloned genes encoding sulfotransferases from fruit fly (Drosophila melanogaster), human (Homo sapiens), rat (Rattus norvegicus), chicken (Gallus gallus), canine (Canis lupus), porcine (Sus scrofa) and equine (Equus caballus) using liver cDNA preparations. We further synthesized genes encoding sulfotransferases from the model organisms Zebrafish (Danio rerio) and the nematode Caenorhabditis elegans. Only a few bacterial PAPS-dependent sulfotransferases have been described, e.g., for the sulfation of nodular rod factors[15]. Recently, PAPS-dependent sulfation of aromatic compounds was also described in bacteria[16], and we therefore also examined a few bacterial genes that are only distantly related to the plant or animal genes (Fig. 2a).

These additional genes were similarly expressed in E. coli. Given that phenol sulfotransferases may have activity towards several compounds, we examined the activity of all 40 cloned sulfotransferases against p-coumaric acid, as well as the polyphenol resveratrol and the flavonoid kaempferol, which are all of interest as biotechnological products[17–19]. In addition, we tested vanillic acid, a small phenolic acid. After overnight growth, culture supernatants were isolated and examined by HPLC for the depletion of substrate (Fig. 2b) and the occurrence of a respective product (Fig. 2c). The results demonstrate that a sulfotransferase was required for the conversion of all four substrates, and that all compounds could act as a substrate for at least one of the selected enzymes under the screening conditions. For the expected product of p-coumaric acid, zosteric acid, the product matched a chemically synthesized standard. For resveratrol, kaempferol and vanillic acid, emerging product peaks that matched the expected shift in retention time relative to the substrate were identified when comparing to the chromatogram of a control

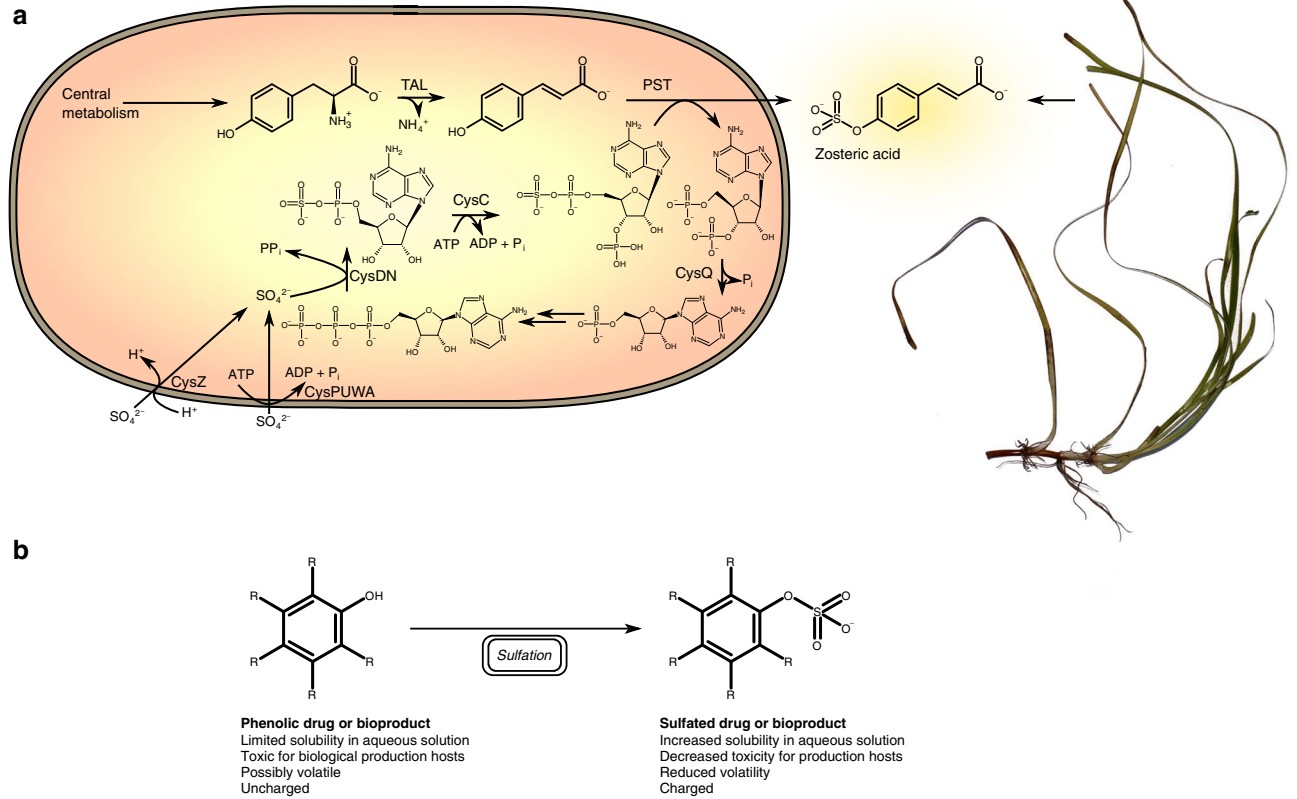

**Fig. 1** The zosteric acid microbial cell factory. **a** The optimized cell factory for producing sulfated biochemicals is exemplified by the production of zosteric acid, a plant biochemical from eelgrass (right side). The cell factory expresses a tyrosine ammonia-lyase (TAL) for deamination of tyrosine, a phenol sulftransferase (PST) for O-sulfonation of p-coumaric acid, and overexpresses uptake mechanisms for sulfate (here either CysZ or CysPUWA) as well as enzymes for activation of sulfate into 3′-phosphoadenosine 5′-phosphosulfate by CysDN and CysQ, and hydrolysis of 3′-phosphoadenosine 5′-phosphate (CysQ). **b** Sulfation as a general process for modifying a drug or a biologically produced compound for enhanced properties

strain (Supplementary Fig. 2). A total of 18 of the 40 tested genes encoding sulfotransferases resulted in product formation. None of the genes from *Z. marina*, nor the *A. thaliana* gene resulted in measurable product, while 17 of the positives were sequences from animal origin, and the product of a single bacterial gene, *H. ochraceum* DSM 14365 Hoch_6098, resulted in product formation from kaempferol and to a lesser extent resveratrol.

**A range of compounds may be subjects to sulfation**. As the active enzymes had shown differences in activity toward four selected compounds, we further explored the substrate specificity of *E. coli* strains expressing a number of selected representative enzymes: SULT1A1 from *Rattus norvegicus*, SULT1B1, SULT1C1 and SULT1E1 from *Gallus gallus domesticus*, SULT1ST1 and SULT6B1 from *D. rerio* and Hoch_6098 from *H. ochraceum*. The strains were grown in minimal media supplemented with either ferulic acid, 3-hydroxy-4-methoxy-cinnamic acid, 4-acet-amidophenol, naringenin, 4-vinylphenol, 4-ethylphenol, 4-ethyl-guaiacol, 4-nitrophenol or 4-methylumbelliferone (4-MU). For all 13 tested compounds, at least one sulfotransferase was able to act upon it, and several enzymes also demonstrated a high degree of promiscuity (Fig. 2d). This demonstrates that a wide range of sulfated phenolic compounds can be generated using microbial cell factories.

**Zosteric acid can be produced in *E. coli* and yeast**. Zosteric acid is an attractive biochemical for production by itself because of its antifouling properties, and it may also be used as building block with interesting properties due to its charged sulfate group. Additionally, it may be a preferred intermediate in p-coumaric

acid production as it is less toxic to the production organism. The growth of *E. coli* in minimal media is inhibited at increasing concentrations of p-coumaric acid, while this is not the case for zosteric acid (Supplementary Fig. 3). It would therefore be possible to accumulate the non-toxic zosteric acid in the fermentation broth to high concentrations and later on convert it to p-coumaric acid. Zosteric acid was shown to be stable in fermentation media, and in the presence of *E. coli* (Supplementary Fig. 4).

Among the screened enzymes, SULT1A1$_{Rno}$ showed the greatest production of zosteric acid from p-coumaric acid, and we therefore chose this enzyme for creating a zosteric acid microbial cell factory. p-coumaric acid may be formed in the host organism from tyrosine by non-oxidative deamination catalyzed by a tyrosine ammonia-lyase (TAL). Thus, we combined the gene encoding SULT1A1$_{Rno}$ with three different TAL-encoding genes: the commonly used tyrosine ammonia-lyases from *Rhodobacter sphaeroides* (TAL$_{Rsp}$) and *Rhodobacter capsulatus* (TAL$_{Rca}$) as well as the recently described tyrosine ammonia-lyase from *Flavobacterium johnsoniae* (TAL$_{Fjo}$)[9] in *E. coli* BL21(DE3). The resulting strains were grown in minimal medium with or without supplementation of tyrosine. In addition, we tested different levels of induction of the expression system. It was clear that expression of a TAL-encoding gene was required for the production of pHCA, and that this enabled the cells to further produce ZA (Fig. 3). Although the resulting titer of p-coumaric acid could be increased by the higher induction level, this was found to be detrimental to the titer of zosteric acid. Furthermore, supplementing the medium with 2 mM tyrosine allowed the strains to reach even higher titers of p-coumaric acid, while the titer of zosteric acid was largely unaffected. The highest titers of zosteric

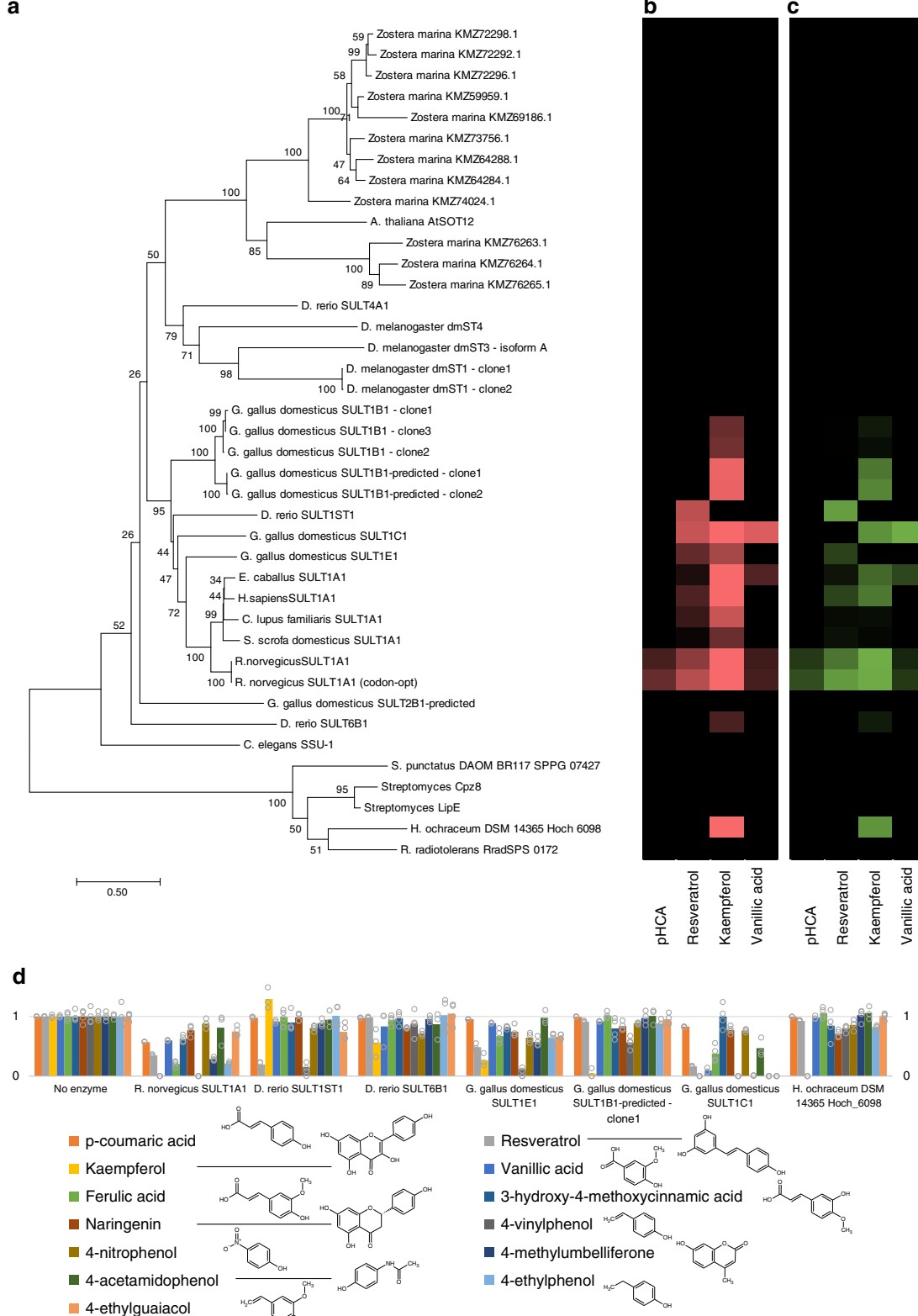

**Fig. 2** Activity and relationship between screened sulfotransferases. The phylogenetic relationship **a** of the chosen sulfotransferases shown together with **b** the consumption of substrates (pHCA (*p*-coumaric acid), resveratrol, kaempferol, and vanillic acid) and appearance (**c**) of product peaks. **d** The substrate specificity of selected phenol sulfotransferases when the production organisms were challenged with selected compounds (*p*-coumaric acid, orange; resveratrol, light gray; kaempferol, yellow; vanillic acid, blue; ferulic acid, light green; 3-hydroxy-4-methoxycinnamic acid, navy blue; naringenin, rust red; 4-vinylphenol, dark gray; 4-nitrophenol, golden brown; 4-methylumbelliferone, dark blue; 4-acetamidophenol, dark green; 4-ethylphenol, light blue; 4-ethylguaiacol, light brown). The height of the bars show the relative remaining amount of compounds relative to a strain that harbors an empty expression plasmid shown (as averages of individual points plotted in circles). Source data of Fig. 2d are provided as a source data file

| | | | No addition | | | | | | 2 mM tyrosine | | | | | |
| | | | 0 IPTG | | 0.1 mM IPTG | | 1 mM IPTG | | 0 IPTG | | 0.1 mM IPTG | | 1 mM IPTG | |
| Strain | TAL | PST | pHCA | ZA | pHCA | ZA | pHCA | ZA | pHCA | ZA | pHCA | ZA | pHCA | ZA |
|---|---|---|---|---|---|---|---|---|---|---|---|---|---|---|
| CBJ1013 | None | SULT1A1$_{Rno}$ | 0 ± 0 | 0 ± 0 | 0 ± 0 | 0 ± 0 | 0 ± 0 | 0 ± 0 | 0 ± 0 | 0 ± 0 | 0 ± 0 | 0 ± 0 | 0 ± 0 | 0 ± 0 |
| CBJ1014 | TAL$_{Rsp}$ | SULT1A1$_{Rno}$ | 22 ± 5 | 1 ± 1 | 46 ± 2 | 44 ± 2 | 61 ± 7 | 27 ± 3 | 61 ± 14 | 0 ± 0 | 344 ± 33 | 52 ± 3 | 362 ± 32 | 9 ± 4 |
| CBJ1015 | TAL$_{Rca}$ | SULT1A1$_{Rno}$ | 4 ± 0 | 0 ± 0 | 41 ± 1 | 78 ± 2 | 74 ± 2 | 41 ± 3 | 4 ± 0 | 0 ± 0 | 510 ± 1 | 113 ± 4 | 431 ± 5 | 30 ± 2 |
| CBJ1016 | TAL$_{Fjo}$ | SULT1A1$_{Rno}$ | 79 ± 3 | 2 ± 1 | 164 ± 7 | 187 ± 11 | 402 ± 19 | 88 ± 14 | 222 ± 16 | 1 ± 0 | 1543 ± 16 | 165 ± 7 | 1566 ± 37 | 88 ± 17 |
| CBJ1246 | TAL$_{Fjo}$ | None | 82 ± 17 | 0 ± 0 | 308 ± 15 | 0 ± 0 | 480 ± 17 | 0 ± 0 | 275 ± 194 | 0 ± 0 | 1677 ± 42 | 0 ± 0 | 1617 ± 21 | 0 ± 0 |

**Fig. 3** Production of zosteric acid directly from glucose in minimal medium in recombinant *E. coli*. Selected strains expressing SULT1A1$_{Rno}$ in combination with either no tyrosine ammonia-lyase (TAL) or one of the tyrosine ammonia-lyases from *Rhodobacter sphaeroides* (TAL$_{Rsp}$), *Rhodobacter capsulatus* (TAL$_{Rca}$), or *Flavobacterium johnsoniae* (TAL$_{Fjo}$), respectively, were grown in M9 medium with 0.2% glucose for 24 h. Resulting production titers (µM) of *p*-coumaric acid (pHCA) or zosteric acid (ZA) and standard deviations (± for *n* = 3) are shown. Source data are provided as a Source Data file

acid were reached in the strain combining SULT1A1$_{Rno}$ with TAL$_{Fjo}$—both under induction with 0.1 mM ITPG and 1 mM IPTG. The growth was significantly influenced by the induction with IPTG; and while it mainly affected the growth rate at 0.1 mM IPTG, it increased the lag phase at 1 mM (Supplementary Fig. 5).

In order to establish whether these results are limited to a bacterial host, we similarly expressed the TAL and PST genes in *Saccharomyces cerevisiae* (Supplementary Table 1). Expression of SULT1A1$_{Rno}$ alone enabled the production of ZA from supplemented pHCA, and in combination with the expression of TAL$_{Fjo}$, *p*-coumaric acid and zosteric acid (13 µM ZA) were produced in minimal media with glucose. As the production was higher for *E. coli*, we chose this organism for further studies.

**Optimization of activated sulfate reaction**. Zosteric acid was only formed, when there was a supply of exogenous *p*-coumaric acid or if the cells were capable of synthesizing *p*-coumaric acid. However, increased tyrosine supply, which resulted in increased concentrations of *p*-coumaric acid, did not result in increasing amounts of zosteric acid being formed. In fact, increased *p*-coumaric acid concentration could even decrease the amount of zosteric acid formed, and similarly, increasing the concentration of sulfate in the medium had no positive effect (Fig. 4), indicating that an alternative bottleneck was found in the production pathway.

The immediate substrate for sulfation is PAPS, a sulfate- and ATP-derived nucleotide, so we attempted to overproduce PAPS by overexpressing the *E. coli* genes encoding sulfate adenylyl-transferase (*cysDN*), and adenosine 5′-phosphosulfate kinase (*cysC*) from the assimilatory sulfate pathway on a separate plasmid. This overexpression resulted in a 36% increase in ZA produced from *p*-coumaric acid in *E. coli* expressing SULT1A1$_{Rno}$ (Fig. 4). It is however known that the product of the PAPS-dependent sulfation reactions, 3′-phosphoadenosine 5′-mono-phosphate (PAP) is inhibitory to the sulfotransferase reaction in vitro[20], thereby conferring a regulatory role of PAP in various organisms[21]. CysQ has initially been suggested to control the amount of PAPS[22], but was found to hydrolyze PAP in vitro[23]. We therefore overexpressed CysQ both alone and from an artificial operon, merging the *cysQ* gene and its native RBS to the end of the *cysDNC* operon. While *cysQ* alone was not sufficient, the combination with *cysDNC* dramatically improved the titers of zosteric acid produced to 2.2–2.3 mM.

**Improved uptake of sulfate enhances zosteric acid production**. To further investigate potential ways to improve the production of ZA, we conducted a transcriptional analysis of CBJ1041 overexpressing SULT1A1$_{Rno}$ and the artificial *cysDNCQ* operon relative to a control strain, CBJ1055 by RNA-seq. The results indicated that the production strain was starved for sulfate (Table 1). Besides *cysDNCQ*, which are intentionally overexpressed, the most upregulated genes were *sbp* and genes belonging to the *ssuEADCB* operon, where *ssuA*, *ssuB* and *ssuC* encode an aliphatic sulfonate ABC transporter, and *ssuD* and *ssuE* encode a FMNH$_2$-dependent alkanesulfonate monooxygenase and a NADPH-dependent FMN reductase, respectively. The *ssuEABCD* operon is suppressed when sulfate is present, and is regulated by CysB and Cbl (CysB-like regulator, 2.1-fold upregulated itself)[24]. The *cbl* expression is dependent on CysB[25], and APS is an antagonist to Cbl-mediated activation[26]. *sbp* encodes a periplasmic protein that allow the CysPAWU sulfate ATP-binding cassette (ABC) transporter[27,28] to function without CysP[29]. Among repressed genes were the transporter subunit encoded by *livH*, part of the *livKHMGF* operon regulated by the leucine-responsive protein (Lrp)[30], the periplasmic protein encoded by *malM*, and also *nuoK*, coding for an inner membrane component of NADH dehydrogenase, but none of the other genes from the *nuo* operon were regulated by 1.5-fold or more. Gene Ontology Enrichment Analysis demonstrated that among genes with *p*-value < 0.05, enriched biological processes were the processes of transport, and in particular transport of the amino acids leucine, isoleucine, valine, phenylalanine and dipeptides as wells as "sulfur compound metabolic process". The results also indicate that overexpression of the SULT1A1$_{Rno}$ in combination with *cysDNCQ* results in sulfate starvation.

**Improved production by optimizing sulfate uptake**. Since higher concentrations of sulfate in the growth medium did not to further improve production (Fig. 4), we hypothesized that sulfate transport may be a limiting step in the production of zosteric acid. Sulfate can be transported across the cell membrane in bacteria by proteins belonging to several families as reviewed by Aguilar-Barajas et al.[31]. We therefore constructed several strains overexpressing different uptake systems including the native proton-symporter CysZ[32,33], as well as the CysPUWA ABC-transporter. We additionally constructed an artificial operon, replacing the thiosulfate-preferring subunit encoded by *cysP* with the sulfate-preferring subunit encoded by *sbp*. Finally, we also examined CysP from *Bacillus subtilis* (CysP$_{Bsu}$), which belongs to the inorganic phosphate transporter (PiT) family, since it was previously found to be able to restore a sulfate starving phenotype of a *cysP sbp* double mutation in *E. coli*[34].

The transporters encoded by *cysPUWA* and *cysP$_{Bsu}$* both improved the titer of zosteric acid, when expressed in combination with *cysDNCQ*, most notably by 3.5-fold under high concentrations of *p*-coumaric acid (Fig. 4). Expression of the transporters was however not consistently positive and could even hamper the production of zosteric acid. Considering the detrimental effects of overexpression of membrane proteins, we tested whether expression from a low-copy plasmid of two transporters would be beneficial. Indeed, the resulting strains (CBJ1242 and CBJ1255) significantly improved the titers of zosteric acid over the high-copy plasmids, and the for low-copy

| Transporter | CysDNC | Sulfotransferase | CysQ | ZA | ZA | ZA | ZA |
|---|---|---|---|---|---|---|---|
| – | – | – | – | 0 ± 0 | 0 ± 0 | 0 ± 0 | 0 ± 0 |
| – | – | SULT1A1$_{Rno}$ | – | 700 ± 125 | 678 ± 127 | 331 ± 44 | 320 ± 37 |
| – | CysDNC | SULT1A1$_{Rno}$ | – | 790 ± 49 | 924 ± 65 | 452 ± 62 | 484 ± 10 |
| – | – | SULT1A1$_{Rno}$ | CysQ | 653 ± 29 | 581 ± 22 | 242 ± 5 | 223 ± 7 |
| – | CysDNC | SULT1A1$_{Rno}$ | CysQ | 2244 ± 271 | 2324 ± 268 | 810 ± 18 | 913 ± 27 |
| – | CysDNC | – | CysQ | 0 ± 0 | 0 ± 0 | 0 ± 0 | 0 ± 0 |
| CysZ | – | SULT1A1$_{Rno}$ | – | 255 ± 41 | 314 ± 74 | 215 ± 33 | 209 ± 26 |
| CysPUWA | – | SULT1A1$_{Rno}$ | – | 590 ± 170 | 616 ± 225 | 317 ± 196 | 280 ± 160 |
| CysP$_{Bsu}$ | – | SULT1A1$_{Rno}$ | – | 500 ± 178 | 501 ± 195 | 329 ± 162 | 322 ± 129 |
| Sbp-CysUWA | – | SULT1A1$_{Rno}$ | – | 411 ± 263 | 384 ± 277 | 205 ± 144 | 192 ± 143 |
| CysZ | CysDNC | SULT1A1$_{Rno}$ | CysQ | 728 ± 78 | 912 ± 223 | 580 ± 128 | 533 ± 97 |
| CysPUWA | CysDNC | SULT1A1$_{Rno}$ | CysQ | 2527 ± 300 | 2827 ± 370 | 1153 ± 233 | 1048 ± 64 |
| CysP$_{Bsu}$ | CysDNC | SULT1A1$_{Rno}$ | CysQ | 2892 ± 15 | 2702 ± 194 | 3713 ± 354 | 4148 ± 739 |
| Sbp-CysUWA | CysDNC | SULT1A1$_{Rno}$ | CysQ | 1747 ± 35 | 1859 ± 116 | 701 ± 115 | 744 ± 178 |
| CysZ (low copy) | CysDNC | SULT1A1$_{Rno}$ | CysQ | 1670 ± 45 | 1681 ± 83 | 554 ± 12 | 526 ± 10 |
| CysPUWA (low copy) | CysDNC | SULT1A1$_{Rno}$ | CysQ | 3027 ± 32 | 3115 ± 69 | 2271 ± 32 | 2695 ± 77 |

**Fig. 4** Improving the supply of activated sulfate promotes production of sulfated products. Strains expressing SULT1A1$_{Rno}$ from pETDuet-1 alone or with in combination with either an empty pRSFDuet-1 plasmid, a derived plasmid harboring the *cysDNC* operon, *cysQ*, a synthetic *cysDNCQ* operon, various sulfate transport genes or combinations thereof were grown in M9 medium with either low or high concentration of *p*-coumaric acid (pHCA) and potassium sulfate for 24 h in 96-well plates. Titers (μM) of produced zosteric acid (ZA) are shown together with standard deviations (± for $n = 3$). No production of ZA occurred in the absence of SULT1A1$_{Rno}$. Source data are provided as a Source Data file

---

**Table 1 Transcriptomic response to induction of zosteric acid production**

| Gene | Fold change | FDR *p*-value | Gene function |
|---|---|---|---|
| *cysQ* | 43.5 | 1.12E−68 | Adenosine-3′(2′),5′-bisphosphate nucleotidase |
| *cysC* | 10.1 | 4.32E−21 | adenylylsulfate kinase |
| *cysD* | 6.5 | 4.10E−09 | Sulfate adenylyltransferase, CysD subunit |
| *cysN* | 4.9 | 1.30E−07 | Sulfate adenylyltransferase, CysN subunit |
| *ssuA* | 6.5 | 4.24E−07 | Aliphatic sulfonate ABC transporter - periplasmic binding protein |
| *livH* | −2.7 | 2.12E−06 | Branched chain amino acid and phenylalanine transporter - membrane subunit |
| *ssuE* | 3.7 | 2.12E−04 | NADPH-dependent FMN reductase |
| *sbp* | 2.4 | 1.53E−02 | Sulfate/thiosulfate ABC transporter - periplasmic binding protein Sbp |
| *cbl* | 2.1 | 1.62E−02 | Cbl DNA-binding transcriptional activator |
| *malM* | −2.2 | 1.62E−02 | Maltose regulon periplasmic protein |
| *nuoK* | −2.2 | 4.80E−02 | NADH:ubiquinone oxidoreductase, membrane subunit K |

Pairwise comparison of zosteric acid producing strain relative to control. Genes with false-discovery rate (FDR)-corrected *p*-values (*t*-test) smaller than 0.05

---

*cysPUWA*, the titers were improved by roughly 35%. While high pHCA concentrations generally yielded lower zosteric acid titers, the best conditions were reached by combining the expression of the transporter encoded by *cysP$_{Bsu}$* with high *p*-coumaric acid and sulfate concentration, resulting in 4.1 mM (1 g L$^{−1}$) zosteric acid being produced. For all other strains, the high *p*-coumaric acid concentration was negatively influencing the amount of zosteric acid produced. Repeating the experiments in shake flasks rather than in deep-well plates, gave similar results, as CBJ1255 (SULT1A1$_{Rno}$, CysPUWA$_{low\ copy}$, CysDNCQ) and CBJ1258 (SULT1A1$_{Rno}$, CysP$_{Bsu}$, CysDNCQ) reached 2.81 ± 0.33 mM (±standard deviation for $n = 3$) and 2.81 ± 0.48 mM zosteric acid from M9 with high level of *p*-coumaric acid and sulfate, respectively.

Overexpression of the sulfate uptake and activation genes were then combined with expression of the phenol sulfotransferase and tyrosine ammonia-lyase for complete synthesis of zosteric acid from tyrosine or even glucose (Fig. 1 and Table 2). As previously shown, co-expression of TAL$_{Fjo}$ and SULT1A1$_{Rno}$ resulted in production of both *p*-coumaric acid and zosteric acid. Over-expression of sulfate uptake and activation enzymes improved the conversion to the latter, reaching 2.56 mM zosteric acid. Fed batch fermentation of a strain overexpressing SULT1A1$_{Rno}$, TAL$_{Fjo}$, CysP$_{Bsu}$ and CysDNCQ, resulted in a zosteric acid titer of 3.2 mM (0.78 g L$^{−1}$) from minimal salts media, and 20.4 mM (5.0 g L$^{−1}$) with supplemented tyrosine (Fig. 5). The specific production reached 0.15 g L$^{−1}$ OD$_{600}$$^{−1}$. Acetate was observed to accumulate during the early part of the fermentation, likely

**Table 2 Production of zosteric acid from glucose or tyrosine**

| Strain | TAL | PST | Transport | Activation | Supplement | pHCA /mM | ZA /mM |
|---|---|---|---|---|---|---|---|
| CBJ1262 | TAL$_{Fjo}$ | SULT1A1$_{Rno}$ | None | None | 10 mM tyrosine | 3.47 ± 0.98 | 0.29 ± 0.16 |
| CBJ1264 | TAL$_{Fjo}$ | SULT1A1$_{Rno}$ | None | CysDNCQ | 10 mM tyrosine | 1.61 ± 0.5 | 2.23 ± 0.94 |
| CBJ1266 | TAL$_{Fjo}$ | SULT1A1$_{Rno}$ | CysPUWA | CysDNCQ | None | 0.52 ± 0.21 | 0.25 ± 0.09 |
| | | | | | 2 mM tyrosine | 0.40 ± 0.26 | 1.38 ± 0.23 |
| | | | | | 10 mM tyrosine | 1.00 ± 0.44 | 2.56 ± 0.87 |

Resulting production titers (μM) of *p*-coumaric acid (pHCA) or zosteric acid (ZA) are shown as averages with standard deviations (± for *n* = 3). Source data are provided as a Source Data file

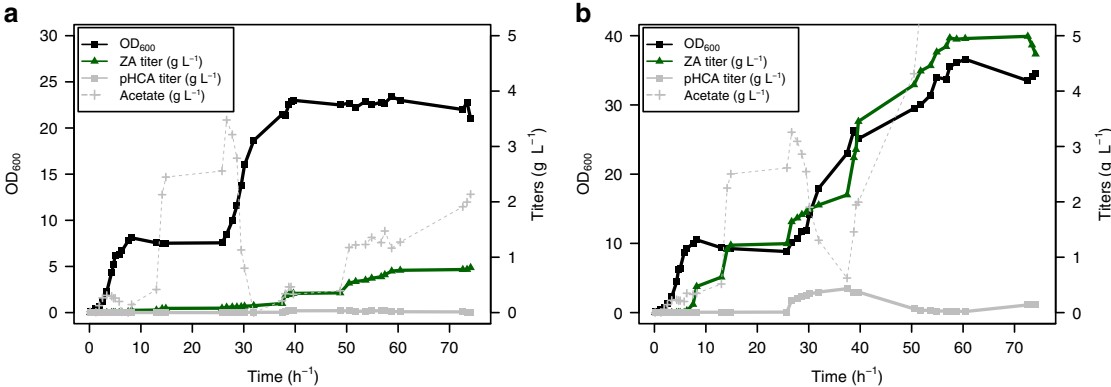

**Fig. 5** Production of zosteric acid in fed-batch bioreactor cultivation. Growth and production in a 1-L fermenter under fed-batch conditions in minimal salts media without tyrosine (**a**) or with supplementation of exogenous tyrosine (**b**)

resulting in inhibition of growth. The results suggest that with the optimized strains, the supply of tyrosine was the limiting factor. Future work could focus on combination of the presented work with high-yield conversion of glucose to tyrosine by means previously reported[35].

## Discussion

Here, we present a general process for high-titer production of sulfated biochemicals through screening of sulfotransferases and optimization of sulfate uptake and activation. We link a biosynthetic pathway to a phenolic compound with phenol sulfotransferase activity in a microbial cell factory. This enabled the complete biosynthesis of a sulfated phenolic compound, zosteric acid, from a minimal medium with glucose as a carbon source. Conclusively, the sulfated product can be formed from an unsulfated precursor molecule when this is either produced in vivo or supplemented exogenously. An efficient precursor supply is required, as exemplified by the various tyrosine ammonia-lyases and *p*-coumaric acid levels tested, however at excessive substrate levels, there is inhibition of product formation which is consistent with formation of a dead-end complex with PAP-bound enzyme[36]. We demonstrate that the sulfate donor PAPS and sulfate itself becomes a limiting factor for the production of sulfated biochemicals. Therefore, we examined overexpression of genes encoding sulfate uptake, activation and recycling of nucleotides, which combined forms a sulfate assimilation cycle. We found that for production of a sulfated biochemical all of these reactions were required for the highest production titers. The sulfate pathways are naturally regulated at the genetic level in response to the availability of sulfate assimilation metabolites, however, this regulation proved insufficient in a strain that was genetically modified to produce large amounts of a sulfated biochemical. The exemplary product, zosteric acid, was found in the supernatant of bacterial cultures, suggesting that it is secreted from the cell. We have future research interest in identifying the possible transport mechanisms for this and other sulfated compounds.

Synthesis of sulfated phenolic compounds using enzymes enables specific sulfation of selected groups using a sulfate carrier as donor through the action of either bacterial arylsulfate sulfotransferases[37] or PAPS-dependent sulfotransferases[38]. As a biocatalytic process, this is limited by the supply of excess sulfate donors. These could come in the form of PAPS[39] or an existing phenol–sulfate conjugate such as *p*-nitrophenol sulfate (*p*-NPS)[40]. Using the metabolic capabilities of a whole-cell factory, the obstacle of having a specific sulfur source is readily overcome.

In the present work, we chose to perform the screening of sulfotransferase candidates in vivo, even though *p*-NPS or 4-methylumbelliferyl sulfate (MUS)[41] could be used in enzymatic assays due to the generation of color and fluorescence of their respective non-sulfated counterparts, *p*-nitrophenol and 4-methylumbelliferone. Both compounds were included in this study as substrates for sulfation, but as shown in Fig. 2, the various enzymes did not have identical preference for these compounds, arguing for using in vivo rather than in vitro screening for the sulfotransferases suitable to be used in a microbial cell factory.

Previously characterized PAPS-dependent phenol sulfotransferases are predominantly mammalian, and their natural function is the clearance of exogenous compounds, since sulfate conjugates are less toxic and more soluble, thereby facilitating their excretion. The enzymes also function to activate and stabilize hormones. Here, we have included bacterial enzymes as well. The sulfation of an aminoribose-moiety of a secondary metabolite in *Streptomyces* occurs through a sulfation of an intermediate: a triketide pyrone. The reaction is catalyzed by a PAPS-dependent sulfotransferase Cpz8, and subsequent transfer of the sulfate group by a PAPS-independent sulfotransferase Cpz4 to the aminoribose-moiety of an intermediate in the synthesis of sulfated caprazamycin antibiotics[16,42]. A similar transfer of

sulfate from a donor molecule rather than PAPS occurs in the biosynthesis of similar fatty acid nucleoside antibiotics[43]. We included the codon optimized cpz8 gene as well as the homolog lipE[43], and the closest homolog to Cpz8 in eukarya; namely the gene encoded by SPPG_07427 in the fungus Spizellomyces punctatus DAOM BR117. We also included the Haliangium ochraceum DSM 14365 Hoch_6098 and Rubrobacter radiotolerans RradSPS_0172, the closest homologs to SPPG_07427 by BLASTP at the time of searching. Interestingly, Hoch_6098 showed reactivity towards resveratrol, and thus appears to be a functional bacterial PAPS-dependent phenol sulfotransferase.

Producing sulfated conjugates as end-products in biotechnology has the potential to overcome the problems of toxic effects of the non-conjugated products. For example, p-coumaric acid is highly toxic to E. coli, making it challenging to produce high titers without two-phase fermentations[44]. Likewise, polyphenolic compounds are difficult to produce in large amount due to various inhibitory effects. While polyphenols are for example reported to have anti-cancer effects through targeting mitochondrial ATPases, they also target the ATPase of E. coli[45], a biotechnological production organism which has been employed for the production of flavonoids and resveratrol. Also, the end-product could be subject to either chemical degradation, or catabolism of the host organism. To circumvent these issues, formation of a sulfate conjugate could protect both the end-product and the production host. This study offers guidance to the further development of microbial cell factories for production of a wide range of sulfated compounds as final products or as soluble, non-toxic derivatives.

## Methods

**Chemicals, strains and media**. All chemicals were purchased from Sigma-Aldrich, except trans-resveratrol 3-sulfate sodium salt (Santa Cruz Biotechnology, USA) and zosteric acid (ChiroBlock, Germany). E. coli strains were routinely grown in rich medium, 2xYT or LB, with appropriate antibiotics for selection of plamids; 34 μg mL$^{-1}$ chloramphenicol for pLysS, 100 μg mL$^{-1}$ ampicillin for pETDuet-1-derived plasmids, 50 μg mL$^{-1}$ spectinomycin for pCDFDuet-1 derived plasmids, and 50 μg mL$^{-1}$ kanamycin for pRSFDuet-1-derived plasmids. For growth experiments, M9 minimal medium with 0.2% glucose and appropriate antibiotics were used. S. cerevisiae strains were grown in SC – ura media for selection of plasmids and for growth experiments in a defined minimal medium[46] based on Delft medium[47] with 25 mg L$^{-1}$ histidine and 75 mg L$^{-1}$ leucine, which has been used in similar experiments[9].

**Gene identification and cloning**. DNA fragments encoding SSU-1$_{Ceb}$, SULT1ST1$_{Dre}$, SULT4A1$_{Dre}$, SULT6B1$_{Dre}$, AtSOT12, Cpz8, LipE, SPPG_07427, Hoch_6098 and RradSPS_0172, SULT1A1$_{Rno}$ and 12 predicted sulfotransferase genes from Zostera marina[11] were synthesized (GeneArt/Life Technologies), so that they carried 5′ (TAGAAATAATTTTGTTTAACTTTAAGAAGGAGATA TACC) and 3′ (CAAGCTTGCGGCCGCATAATGCTTA) flanking regions and were codon optimized for expression in E. coli (Supplementary Data 1). The flanking regions allowed integration in the plasmid pETDuet-1 (Invitrogen/Life Technologies) digested by restriction enzymes NcoI and SalI using either Gibson Assembly Mastermix or NEBuilder isothermal assembly (New England Biolabs). Reaction products were transformed into chemically competent E. coli NEB5-alpha (New England Biolabs) or KRX (Promega), selecting for resistance to 100 μg mL$^{-1}$ ampicillin. Similarly, sulfotransferase genes were cloned from cDNA (Zyagen, San Diego, USA) using the oligos listed in Supplementary Data 2. Correct inserts were verified by Sanger sequencing and final plasmids are listed in Supplementary Data 3. Constructs were tested for activity in E. coli expression strains KRX or BL21 (DE3)pLysS (Invitrogen). For plasmid pCBJ271, the cysDNC cluster was amplified by PCR from E. coli MG1655 chromosomal DNA using primers CBJP491 and CBJP492 shown in Supplementary Data 2. The plasmid pRSFDuet-1 (Life Technologies) was digested by the restriction endonucleases NdeI and BglII. The gene cluster was inserted into the digested plasmid using the Gibson reaction (New England Biolabs), resulting in plasmid pCBJ271. For cloning of cysQ, the gene was amplified from E. coli MG1655 chromosomal DNA using primers CBJP495 and CBJP496 and inserted into pETDuet-1 digested with NdeI and BglII, in a Gibson reaction, resulting in pCBJ269. pCBJ434 was made by transferring the NdeI-XhoI fragment with cysQ was transferred to pRSFDuet-1 using restriction-ligation. For the combined expression of cysDNC and cysQ in an artificial operon, cysDNCQ, the two parts were amplified by PCR from E. coli MG1655 chromosomal DNA using

the primer pairs, CBJP491 + CBJ497 and CBJP496 + CBJP498, respectively, and thereby linking cysC with the sequence starting 35 nucleotides upstream of cysQ. Again, the parts were inserted into the digested vector, resulting in plasmid pCBJ272.

Genes encoding transport proteins were amplified from E. coli MG1655 using primer pairs CBJP891 + CBJP892 (cysZ) or CBJP893 + CBJP894 (cysPUWA) or from Bacillus subtilis 168 using CBJP912 + CBJP913 (cysP$_{Bsu}$). Also the cysP was replaced in cysPUWA with sbp, by overlap-extension PCR using primers CBJP910 + CBJP894 on the two products made by primer pair CBJP911 + CBJP912 (sbp) and CBJP908 + CBJP894 (cysUWA). PCR products and pRSFDuet-1 were digested with HindIII and NotI and ligated, resulting in plasmids pCBJ364, pCBJ365, pCBJ372 and pCBJ373. Similarly, pCBJ333, pCBJ334, pCBJ368 and pCBJ369 were created by insertion into pCBJ272 digested with HindIII and NotI. Also the fragments encoding cysZ and cysPUWA were inserted into pCBJ256 digested with HindIII and NotI, resulting in plasmids pCBJ332 and pCBJ361.

For Saccharomyces cerevisiae strains were constructed in a manner comparable to previously published[9]. Genetic constructs were made in E. coli DH5α growing in LB containing 100 μg mL$^{-1}$ ampicillin for plasmid maintenance. Genes encoding tyrosine ammonia-lyases and phenol sulfotransferase were amplified using the oligonucleotides listed in Supplementary Data 2 and inserted alone or in combination by uracil excision cloning into the vector pCfB132 after the PPGK1 promoter or the PTEF1 promoter, respectively[46]. The finished plasmids (Supplementary Data 3) were transformed into S. cerevisiae CEN.PK102-5B selecting for growth on synthetic dropout medium plates lacking uracil. Stains are listed in Supplementary Data 4.

**Growth conditions**. Screening of phenol sulfotransferase activity: strains containing plasmids were grown overnight in 2xYT with chloramphenicol and ampicillin for selection for pLysS and plasmids encoding phenol sulfotransferases. In order to screen for enzymatic activity, 50 μL of overnight cultures (KRX background) were used to inoculate 950 μL M9 minimal medium with 4 mM MgSO$_4$ in 96-well deep-well plates, containing 0.1 mM IPTG and 0.1% rhamnose for inducing gene expression and either 100 μM pHCA, 100 μM resveratrol, 20 μM kaempferol or 100 μM vanillic acid added from fresh 20 mM stocks in absolute ethanol. The culture grew 24 h at 37 °C in an orbital shaker (300 rpm) before measurement of optical density at 600 nm and sampling of the supernatants. For pHCA and resveratrol 600 μL of the cultures were centrifuged for 10 min at 4000 × g, and subsequently 300 μL of the supernatant was centrifuged again before 200 μL was sampled for HPLC. For kaempferol and vanillic acid, the samples were first mixed with equal volumes of methanol before centrifugation. Experiments were performed in triplicates.

For testing of activation and transport of sulfate, strains were grown as described above, in media with either 2 mM (low) pHCA or 5 mM (high) pHCA, and either 0 mM (low sulfate) or 10 mM (high sulfate) K$_2$SO$_4$ in addition to the MgSO$_4$ present in the M9 media. Evaporation of liquid from the plates over 24 h was around 30–35%.

For testing of inhibitory effect of p-coumaric acid and zosteric acid, E. coli MG1655 was grown in chemically defined M9 minimal media with 0.2% glucose as a carbon source without further addition or with the additions of either 10, 20, 25, 30, 35 or 40 mM p-coumaric acid (pHCA), or with 20 or 40 mM of zosteric acid. All media preparations had been adjusted to pH 7. Cells were grown at 37 °C with agitation and the exponential phase growth rates were determined by following the optical density at 600 nm.

**Transcriptomic analysis**. Strains CBJ1041 (BL21(DE3)/pCBJ256 + pCBJ272) and CBJ1055 (BL21(DE3)/pETDuet-1 + pRSFDuet-1) were grown in 50 mL M9 minimal medium with 0.4% glucose, 2 mM pHCA, 1 mM IPTG, 100 mg mL$^{-1}$ ampicillin and 50 μg mL$^{-1}$ kanamycin in 250-mL baffled shake flasks in a shaking, 37 °C water bath. They were inoculated to OD$_{600}$ = 0.05 from an overnight culture. At OD$_{600}$ = 0.7, cells were harvested for isolation of RNA by mixing 1000 μL with 250 μL ice-cold STOP solution (phenol saturated with 0.1 M citrate buffer, pH 4.3, diluted to 5% in 99% ethanol). 1000 μL-samples were also withdrawn at the same time for HPLC analysis. Samples were kept cold, centrifuged at 6500 × g, 4 °C, for 5 min, followed by removal of supernatant and snap freezing in liquid N$_2$ before storing at −80 °C. The HPLC samples were centrifuged twice at 13,000 × g for 3 min for isolation of supernatant. Cells were lyzed with lysozyme in TE buffer and total RNA was isolated using the RNeasy mini kit (Qiagen). The RNA quality was assessed using RNA NANO prokaryote on a Bioanalyzer (Agilent) and RNA was quantified using a Qubit RNA kit (Thermo Fisher Scientific). Libraries were prepared and run on a NextSeq sequencer (Illumina) using a Mid Output Kit. Paired-end reads were mapped the chromosomal sequence of BL21(DE3) and counted and normalized to fragments per kilobase per million mapped reads (FPKM), before fold-change and false-discovery rate (FDR)-adjusted p-values were calculated for the pairwise comparison. Gene Ontology Enrichment Analysis was performed by a PANTHER Overrepresentation Test, the GO Ontology database released 2019–01–01 using Fisher's exact test with FDR correction[48].

**Fed batch fermentation**. Fermentations were performed in Sartorius 1-L fermenters. Initial batch media (500 mL) contained 2 g L$^{-1}$ KH$_2$PO$_4$, 5 g L$^{-1}$

$(NH_4)_2SO_4$, $2\,g\,L^{-1}$ $MgSO_4\cdot7H_2O$, 2 mL of 1000× M9 vitamin solution, 25 μL of a 1 M $CaCl_2$ solution, $5\,g\,L^{-1}$ glucose, $2\,g\,L^{-1}$ yeast extract (Oxoid LP0021), 4 mL of trace element solution, $100\,\mu g\,mL^{-1}$ ampicillin, $50\,\mu g\,mL^{-1}$ spectinomycin, and $50\,\mu g\,mL^{-1}$ kanamycin. The temperature was adjusted to 37 °C, and pH was adjusted with 2 M NaOH to pH 7.0. 10 mL of an overnight culture of CBJ1292 was used to inoculate the media, and oxygen saturation was kept at minimum 40% dissolved oxygen using a cascade controlling stirring and sparging of air. At $OD_{600} = 4.5$, the cultures were induced with IPTG to 100 μM. At $OD_{600} = 6$, the fed-batch was initiated. The feed contained $2\,g\,L^{-1}$ $KH_2PO4$, $15\,g\,L^{-1}$ $(NH_4)_2SO_4$, $2\,g\,L^{-1}$ $MgSO_4\cdot7H_2O$, 1 mL of 1000× M9 vitamin solution, 25 μL of a 1 M $CaCl_2$ solution, $420\,g\,L^{-1}$ glucose, $2\,g\,L^{-1}$ yeast extract (Oxoid LP0021), 1 mL of trace element solution, $100\,\mu g\,mL^{-1}$ ampicillin, $50\,\mu g\,mL^{-1}$ spectinomycin, $50\,\mu g\,mL^{-1}$ kanamycin and 100 μM IPTG. pH was maintained by automatic adjustment with 2 M NaOH, and manually with slow addition of 1 M HCl. 2–5 mL samples were taken manually for determination of OD, and sampling for glucose, acetate, tyrosine, pHCA and zosteric acid. For the culture with tyrosine addition, multiple additions of tyrosine were added when tyrosine levels were low and little undissolved tyrosine was present after centrifugation.

**Analytics**. HPLC analysis was performed essentially as previously described[9]. Samples were kept cold (4 °C between isolation and analysis). All results are presented as averages and standard deviations of three biological replicates.

**Bioinformatic analysis**. Phylogenetic tree was constructed using muscle alignment[49] of protein sequences and neighbor-joining algorithm[50] with a JTT substitution matrix[51] with 1000 bootstrap replications[52] in MEGA7[53].

**Reporting Summary**. Further information on research design is available in the Nature Research Reporting Summary linked to this article.

## Data availability

Next-generation sequencing data is available at the NCBI Sequence Read Archive (SRA) with the BioProject ID PRJNA478499. Plasmids are available upon request from the laboratory or from Addgene. The source data underlying Figs. 2d, 3 and 4 and Table 2 are provided as a Source Data file.

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

## Acknowledgements

This work was funded by a grant from the Novo Nordisk Foundation (grant number NNF10CC1016517) to the Technical University of Denmark as well as a grant to Christian Bille Jendresen (grant number NNF15OC0015246).

## Author contributions

C.B.J. designed and performed the experiments and wrote the manuscript. A.T.N. supervised the work. Both authors read, corrected and approved the final manuscript.

## Additional information

**Competing interests:** C.B.J. and A.T.N. have filed provisional applications on this work and are co-founders of Cysbio ApS.

