## [Peer Review File · Nature Communications]

Reviewers' comments:

Reviewer #1 (Remarks to the Author):

The authors report the microbial production of zotic acid and sulfated phenolic compounds. The key findings in this work include

- sulfation property of 40 SULTs from various organisms
- some SULTs can react to 17 phenolic compounds
- first example of ZA production using microbe
- strengthen PAPS supplying pathway for improving ZA production

Although the authors' approach is interesting, less impact for the scientists in the field of the material production using microbes; therefore, as described below, I don't think that this manuscript meets criteria of publishing on Nature Communications.

1. Less novelty

The production of sulfated phenolic compounds using *E. coli* has recently been published (Matsumura et al., Scientific Reports 8:7980, 2018). In that paper, sulfated alkaloids were produced using human SULT expressing *E. coli*. So, this work could not "pave the way for a new range of biochemical products with a broad range of applications".

2. Not so difficult

Only one step reaction is required for production of ZA from p-coumaric acid. As the authors known, p-coumaric acid production using *E. coli* has extensively studied; therefore, one step reaction sounds easy.

3. Not so important targets for human life

ZA has bioactivity but is not essential for human life. This point decrease the impact of this manuscript.

My own opinions:

Improvement of sulfate supply is generally difficult. But the authors have succeeded to increase PAPS supply, maybe first successful example?. I think this point is very important for fermentative production research. But the authors do not emphasize the importance of their brilliant results. Although, of course, more detail experiments would be required, I expect the authors to publish their PAPS supply system as soon as possible.

Minors

-Abstract

The should describe improvement of the PAPS supplying pathway.

-L58-60

As described above, I think that this work cannot pave the way for

-L74

As described above, they have already been the subject of biotechnological investigation.

-L90-92

The authors should explain the KRX system more detail.

-Fig. 2

If authors want to show the SULTs can react with various phenolic compounds, authors should confirm which hydroxy group was sulfated. At least tandem MS analysis should be shown.

-Fig. 2

The authors had better show the chemical formula for reader easily understanding.

-Fig. 2d

The color of bars is not easy to be understood. The authors should change the color in an easy-to-

understand manner.

-L117

Why the authors could judge the sulfated resveratrol was 3-sulfate form? How could they eliminate the possibility of 5-sulfate or 4'-sulfate? The authors should explain in the text.

-Supplementary Fig. 3

The growth rate data are generally difficult to reproduce. The authors should show the error-bar and more sampling points in the ZA data would be required.

-Table 1

Did the deleterious effects of IPTG affect to the growth of strains? The growth data should be shown.

-L177-178

I could not find the data of optimization improving p-coumaric acid synthesis. Please specify the data.

-L179-193

I think the yeast data can be eliminated in this manuscript.

-L191

The authors should show the results of products stability. I think that it is important for the practical production.

-L211-213

The data indicate that CysDNC can be dispensable. The authors should confirm whether CysQ is solely sufficient for increasing the production.

-L270

mM?

-Discussion

The PAPS supply system should be discussed!

-L329-333

I think this discussion could be deleted.

-L335-346

Very interesting discussion.

Reviewer #2 (Remarks to the Author):

Jendresen and Nielsen show the production of zosteric acid (the sulfate ester of p-coumaric acid) in *E. coli* and yeast. The results are very nice and clearly presented, and the production of sulfated compounds in microbes is a relevant topic. My major concern is that the study centers almost exclusively on the screening of the engineered strains, but a more detailed biochemical and physiological characterization is missing. Detailed comments are below.

i) The study has a strong screening character. The results from the initial screening of enzymes by endpoint measurements of the product are very nice and led to a good base strain for production of zosteric acid. However, the authors use the same endpoint screening to evaluate further modifications (increasing sulfate transport and conversion to PAP-sulfate). At this point, it would be important to show and discuss biomass specific production rates and growth of the strains. In particular, because it seems that high titers of zosteric acid come with high concentrations of p-coumaric acid. As the authors show p-coumaric is toxic for *E. coli* and the reduced growth may influence productivity, e.g. if the production of zosteric acid is growth coupled. It would strengthen the manuscript to show specific production, substrate consumption and growth rates, ideally in conditions of a bioprocess.

ii) More detailed analysis of the RNAseq data; it is not clear if genes in Table 1 are the only ones upregulated. The authors report upregulation of *cysQ*, *cysC* etc in the production strain, what is obvious since these genes are overexpressed. This should be clarified in the text. In general, the RNAseq data analysis should be more systematic, e.g. testing enrichment of certain regulons or

biological processes, especially those controlled by *csyB* or stress response regulators. Lrp is mentioned in the text but since Lrp has many targets they should all respond (not only the single gene mentioned). Ideally, this analysis would show if cell lack sulfate, as hypothesized in the text, or if additional problems lay within the sulfate assimilation pathway. Maybe cells starve for sulfur-containing amino acids, what explains the Lrp response and would further trigger subsequent stress responses. Moreover, it seems likely that overexpression of *cysQ* removes PAP from the sulfate assimilation pathway, and consequently uptake of sulfate is not the limiting factor.

Minor points:

- Line 61: The pathways and structures in Figure 1 are difficult to read
- Line 179: Has yeast a better tolerance to p-coumaric acid than *E. coli*?
- Line 333: Although the effect of importers was tested, export could be a problem as well
- Line 416: 0 mM sulfate should result in no product formation, was there another source of sulfate.

Reviewer #1 (Remarks to the Author):

The authors report the microbial production of zosteric acid and sulfated phenolic compounds. The key findings in this work include

- sulfation property of 40 SULTs from various organisms
- some SULTs can react to 17 phenolic compounds
- first example of ZA production using microbe
- strengthen PAPS supplying pathway for improving ZA production

Although the authors' approach is interesting, less impact for the scientists in the field of the material production using microbes; therefore, as described below, I don't think that this manuscript meets criteria of publishing on Nature Communications.

1. Less novelty

The production of sulfated phenolic compounds using *E. coli* has recently been published (Matsumura et al., Scientific Reports 8:7980, 2018). In that paper, sulfated alkaloids were produced using human SULT expressing *E. coli*. So, this work could not "pave the way for a new range of biochemical products with a broad range of applications".

Reply: We acknowledge the new Matsumura *et al* publication, and have changed our phrasings accordingly. We do however identify a whole range of novel compounds – and describe how one compound in particular (zosteric acid) can be generated even when no enzymes native to *Zostera marina* were found to be active. We furthermore demonstrate the engineering of a cell factory capable of producing higher quantities of zosteric acid (5 g/L) through novel metabolic engineering of the sulfate pathways.

2. Not so difficult

Only one step reaction is required for production of ZA from p-coumaric acid. As the authors known, p-coumaric acid production using *E. coli* has extensively studied; therefore, one step reaction sounds easy.

Reply: Indeed, one step sounds easy, however, we focus on both the sulfation reactions and co-factor requirements. Also, none of the native enzymes of *Zostera marina* were found to be active, and hence a larger screening was carried out, resulting in the investigation of 40 different primarily uncharacterized sulfotransferases. We analyze the specificity of a these enzymes for a range of substrates. Conclusively, a complete sulfation activation cycle reactions are required for efficient production.

3. Not so important targets for human life

ZA has bioactivity but is not essential for human life. This point decrease the impact of this manuscript.

Reply: Few things are essential for human life, but the societal and environmental impact of a biological antibiofouling agent may indeed be very high. The potential of using the described processes for increasing bioavailability of human nutraceuticals and drugs does have an impact on human life. In the study we have included resveratrol as an example of a nutraceutical, and studies have previously shown increased bioavailability and effectivity of resveratrol-sulfate. Glycosylation is a method widely used in pharmaceutical industry to address solubility of compounds, and we believe that the creation of sulfate esters of phenolic groups is likely to provide an alternative modification approach. We therefore believe that the fundamental description of the class of sulfotransferases, and the optimization of an efficient cell

factory for producing aromatic sulfate esters should be of significant interest also to the pharmaceutical industry.

My own opinions:

Improvement of sulfate supply is generally difficult. But the authors have succeeded to increase PAPS supply, maybe first successful example?. I think this point is very important for fermentative production research. But the authors do not emphasize the importance of their brilliant results. Although, of course, more detail experiments would be required, I expect the authors to publish their PAPS supply system as soon as possible.

Reply: We thank you very much for your positive comment. We have decided to include the PAPS supply study in this manuscript in order to generate a more complete story. We acknowledge the point that the PAPS supply study is highly interesting, and we have therefore changed the abstract to highlight specifically this part.

We have also provided a new panel b in Fig. 1 to reflect the broad applications of the sulfation process.

Minors

-Abstract

The should describe improvement of the PAPS supplying pathway.

Reply: Agree, we have changed the abstract accordingly.

-L58-60

As described above, I think that this work cannot pave the way for

Reply: We acknowledge the work of Matsumura et al and instead highlight the broader scope of our work.

-L74

As described above, they have already been the subject of biotechnological investigation.

Reply: We have rephrased this.

-L90-92

The authors should explain the KRX system more detail.

Reply: We have followed your suggestion.

“We used a KRX strain, for cloning and tight regulation of gene expression from T7 promoters through induction of T7 RNA polymerase by IPTG and rhamnose (Supplementary Fig. 1). The strain combines the T7 RNA polymerase under control of a rhamnose promoter that enables efficient protein overexpression in the absence of recombination and nuclease activities.”

-Fig. 2

If authors want to show the SULTs can react with various phenolic compounds, authors should confirm which hydroxy group was sulfated. At least tandem MS analysis should be shown.

Reply: We can show that the SULTs react with the phenolic compounds, and most of the compounds only have one hydroxyl group subject to sulfation. We believe that distinguishing sulfation patterns is a subject for independent follow-up work with the particular compounds with multiple hydroxyl groups. We have tried tandem MS for several resveratrol products with different retention times. They match resveratrol-3-

sulfate, but may have different retention times. We were unable to distinguish the sulfation patterns, suggesting this is indeed an independent investigation.

-Fig. 2

The authors had better show the chemical formula for reader easily understanding.

Reply: Thank you for the useful comments - We have followed your suggestion and panel d now includes the structures.

-Fig. 2d

The color of bars is not easy to be understood. The authors should change the color in an easy-to-understand manner.

Reply: We have changed the colors for easier identification as suggested.

-L117

Why the authors could judge the sulfated resveratrol was 3-sulfate form? How could they eliminate the possibility of 5-sulfate or 4'-sulfate? The authors should explain in the text.

Reply: A 3- and 5- sulfate would be structurally identical. The different SULT produce different peaks that separate in liquid chromatography, suggesting the SULTs have preferences for different hydroxyl groups. We have changed the text, so we no longer speculate on the identity of the product.

-Supplementary Fig. 3

The growth rate data are generally difficult to reproduce. The authors should show the error-bar and more sampling points in the ZA data would be required.

Reply: In the revised supplementary materials, we have replaced the figure with data from a repetition of the experiment. Individual data points for triplicates are shown.

-Table 1

Did the deleterious effects of IPTG affect to the growth of strains? The growth data should be shown.

Reply: There is indeed a deleterious effect of IPTG induction and we have commented upon this. We have replaced the experimental data with a new table that includes additional points without IPTG induction and have included growth data in the supplementary material (Supplementary Fig. 5).

-L177-178

I could not find the data of optimization improving p-coumaric acid synthesis. Please specify the data.

Reply: This is not important for the remaining part of the manuscript, so we have omitted this part in the revised manuscript.

-L179-193

I think the yeast data can be eliminated in this manuscript.

Reply: While the results may be of interest to the yeast community (resveratrol in for example produced in yeast by the company Evolva), we have removed the section, and included some of the data in the supplementary materials.

-L191

The authors should show the results of products stability. I think that it is important for the practical production.

Reply: This is a very good point. We have now included data on the stability of zosteric acid. It has been fully stable over four weeks in E. coli and yeast cultures, as well as in the media (Supplementary Fig. 4).

-L211-213

The data indicate that CysDNC can be dispensable. The authors should confirm whether CysQ is solely sufficient for increasing the production.

Reply: We have now included experimental data shown in Fig. 3. from a strain only overexpressing *cysQ*, and this confirms that CysQ alone is insufficient for increasing the production.

-L270

mM?

Reply: Correct. Thank you for pointing this out.

-Discussion

The PAPS supply system should be discussed!

Reply: We have included a discussion of this.

“We demonstrate that the sulfate donor PAPS and sulfate itself becomes a limiting factor for the production of sulfated biochemicals. Therefore, we examined over-expression of genes encoding sulfate uptake, activation and recycling of nucleotides, which combined forms a sulfate assimilation cycle. We found that for production of a sulfated biochemical all of these reactions were required for the highest production titers. The sulfate pathways are naturally regulated at the genetic level in response to the availability of sulfate assimilation metabolites, however, this regulation proved insufficient in a strain that was genetically modified to produce large amounts of a sulfated biochemical.”

-L329-333

I think this discussion could be deleted.

Reply: We have followed your suggestion.

-L335-346

Very interesting discussion.

Reply: Thank you. We hope that this emphasizes the importance of the research.

Reviewer #2 (Remarks to the Author):

Jendresen and Nielsen show the production of zosteric acid (the sulfate ester of *p*-coumaric acid) in *E. coli* and yeast. The results are very nice and clearly presented, and the production of sulfated compounds in microbes is a relevant topic. My major concern is that the study centers almost exclusively on the screening of the engineered strains, but a more detailed biochemical and physiological characterization is missing. Detailed comments are below.

i) The study has a strong screening character. The results from the initial screening of enzymes by endpoint measurements of the product are very nice and led to a good base strain for production of zosteric acid. However, the authors use the same endpoint screening to evaluate further modifications (increasing sulfate transport and conversion to PAP-sulfate). At this point, it would be important to show and discuss biomass specific production rates and growth of the strains. In particular, because it seems that high titers of zosteric acid come with high concentrations of *p*-coumaric acid. As the authors show *p*-coumaric is toxic for *E. coli* and the reduced growth may influence productivity, e.g. if the production of zosteric acid is growth coupled. It would strengthen the manuscript to show specific production, substrate consumption and growth rates, ideally in conditions of a bioprocess.

Reply: Thank you very much for your comments. In the revised manuscript we have now included new data from bioreactor cultivations (Fig. 4), addressing some of the concerns. The production of zosteric acid does not inherently come with a high accumulation of *p*-coumaric acid. Rather the conversion to zosteric acid may be seen as being costly in terms of ATP hydrolysis, which is overcome with continuous supply of glucose. The production was shown to be highest with the supplementation of tyrosine, where a specific production is around 0.15 g/L/OD600. It is evident that there is room for improvement of genetic makeup as well as optimization of fermentation conditions, but this, however, we see as an independent follow-up study following well-established routines for increasing tyrosine production beyond the scope of the present manuscript.

ii) More detailed analysis of the RNAseq data; it is not clear if genes in Table 1 are the only ones upregulated. The authors report upregulation of *cysQ*, *cysC* etc in the production strain, what is obvious since these genes are overexpressed. This should be clarified in the text. In general, the RNAseq data analysis should be more systematic, e.g. testing enrichment of certain regulons or biological processes, especially those controlled by *csyB* or stress response regulators. *Lrp* is mentioned in the text but since *Lrp* has many targets they should all respond (not only the single gene mentioned). Ideally, this analysis would show if cells lack sulfate, as hypothesized in the text, or if additional problems lay within the sulfate assimilation pathway. Maybe cells starve for sulfur-containing amino acids, what explains the *Lrp* response and would further trigger subsequent stress responses. Moreover, it seems likely that overexpression of *cysQ* removes PAP from the sulfate assimilation pathway, and consequently uptake of sulfate is not the limiting factor.

This is an interesting discussion. On the individual gene basis we have limited the analysis to the most significantly regulated genes, and at this point in the story, the RNAseq analysis was mainly used to identify the next likely step for improvement rather than a complete description of the intermediate strain. We do recognize the value of the looking into more detail. Examining particular biological processes with gene ontology enrichment analysis, we were able to identify the processes of transport, and in particular transport of the amino acids leucine, isoleucine, valine, phenylalanine and dipeptides as well as "sulfur

compound metabolic process". We have added some text to reflect this and mentioned that the cysDNCQ genes were intentionally over-expressed.

The overexpression of sulfate transporters did improve the production of zosteric acid.

Minor points:

- Line 61: The pathways and structures in Figure 1 are difficult to read

We have increased the fonts and structures sizes. Resolution will be higher in the publication ready figures.

- Line179: Has yeast a better tolerance to p-coumaric acid than E. coli?

Reply: We have followed the suggestion of reviewer 1 and removed the section on yeast.

- Line333: Although the effect of importers was tested, export could be a problem as well

Reply: This is indeed true and we have mentioned this in the discussion.

- Line 416: 0 mM sulfate should result in no product formation, was there another source of sulfate.

Reply: We have corrected the text to make it clear that the K₂SO₄ is in addition to the MgSO₄ present in the M9 media.

REVIEWERS' COMMENTS:

Reviewer #1 (Remarks to the Author):

The authors revised most of the reviewer's previous concerns, and thus the revised paper has been much improved and is in a nice condition now.

I do not have further comments.

Reviewer #2 (Remarks to the Author):

The authors addressed my first point with new experiments and the second point with new data analysis and discussion. All minor points are addressed.

Response to referees

REVIEWERS' COMMENTS:

Reviewer #1 (Remarks to the Author):

The authors revised most of the reviewer's previous concerns, and thus the revised paper has been much improved and is in a nice condition now.
I do not have further comments.

Reviewer #2 (Remarks to the Author):

The authors addressed my first point with new experiments and the second point with new data analysis and discussion. All minor points are addressed.

OUR COMBINED REPLY:

We thank both reviewers for their time and effort in reviewing the manuscript. We are glad that we have satisfactorily addressed the concerns.